# Disease-Induced Modulation of Drug Transporters at the Blood–Brain Barrier Level

**DOI:** 10.3390/ijms22073742

**Published:** 2021-04-03

**Authors:** Sweilem B. Al Rihani, Lucy I. Darakjian, Malavika Deodhar, Pamela Dow, Jacques Turgeon, Veronique Michaud

**Affiliations:** 1Tabula Rasa HealthCare, Precision Pharmacotherapy Research and Development Institute, Orlando, FL 32827, USA; srihani@trhc.com (S.B.A.R.); ldarakjian@trhc.com (L.I.D.); mdeodhar@trhc.com (M.D.); pdow@trhc.com (P.D.); jturgeon@trhc.com (J.T.); 2Faculty of Pharmacy, Université de Montréal, Montreal, QC H3C 3J7, Canada

**Keywords:** the blood–brain barrier, drug transporters, Alzheimer’s disease, stroke, epilepsy, neuroinflammation

## Abstract

The blood–brain barrier (BBB) is a highly selective and restrictive semipermeable network of cells and blood vessel constituents. All components of the neurovascular unit give to the BBB its crucial and protective function, i.e., to regulate homeostasis in the central nervous system (CNS) by removing substances from the endothelial compartment and supplying the brain with nutrients and other endogenous compounds. Many transporters have been identified that play a role in maintaining BBB integrity and homeostasis. As such, the restrictive nature of the BBB provides an obstacle for drug delivery to the CNS. Nevertheless, according to their physicochemical or pharmacological properties, drugs may reach the CNS by passive diffusion or be subjected to putative influx and/or efflux through BBB membrane transporters, allowing or limiting their distribution to the CNS. Drug transporters functionally expressed on various compartments of the BBB involve numerous proteins from either the ATP-binding cassette (ABC) or the solute carrier (SLC) superfamilies. Pathophysiological stressors, age, and age-associated disorders may alter the expression level and functionality of transporter protein elements that modulate drug distribution and accumulation into the brain, namely, drug efficacy and toxicity. This review focuses and sheds light on the influence of inflammatory conditions and diseases such as Alzheimer’s disease, epilepsy, and stroke on the expression and functionality of the BBB drug transporters, the consequential modulation of drug distribution to the brain, and their impact on drug efficacy and toxicity.

## 1. The Blood–Brain Barrier

The blood–brain barrier (BBB) is a complex cellular barrier composed of a tightly sealed monolayer of specialized brain capillary endothelial cells lined on a basement membrane and surrounded by adjacent perivascular astrocytes, pericytes, and microglia. The major function of the BBB is to separate the circulating blood from the brain extracellular fluid and maintain homeostasis in the central nervous system (CNS) (Figure 1).

### 1.1. BBB Facts and Figures

For an average adult, the human brain represents 2% of total body mass. Remarkably, despite this relatively small contribution to total body weight, the brain is composed of ~170 billion neuronal, glial, or non-neuronal cells [1]. Under resting conditions, about 20% of total blood flow from the heart is received by the brain in healthy adults [2,3]. The BBB has a large surface area of almost 15 to 25 m^2^, a total length of capillaries of ~400 miles, microcapillaries with a diameter of 7–10 µm, and an average inter-capillary distance of ~40 µm [1,4,5,6]. These characteristics allow very selective diffusion and penetration of compounds [1,4]. The large BBB surface area and its critical role in maintaining homeostasis of the brain require regulation and support from accompanying brain cells, which all together form the neurovascular unit (NVU).

### 1.2. The Neurovascular Unit

The NVU is a relatively new concept in neuroscience. The NVU was originally described in 1996 as a three-party unit composed of neurons, cerebral blood vessels, and astrocytes, collectively regulating cerebral blood flow [7,8]. In 2001, during the Stroke Progress Review Group meeting of the National Institute of Neurological Disorders and Stroke, endothelial cells were recognized as the NVU blood vessel component to emphasizes the unique relationship between brain cells and cerebral vasculature [9,10]. Since then, the NVU has gained much interest from the neuroscience community and provided a better understanding of the inter-play and signaling between these different cell types. The gate function of the BBB is mainly due to the tightly sealed monolayer of endothelial cells lining the brain capillaries but the interaction of these endothelial cells with surrounding neurons, astrocytes, microglia, and pericytes is key to maintaining proper integrity and function of the BBB [11]. Furthermore, the NVU has become the center of attention for our understanding of the pathology of several neurodegenerative diseases and, consequently, may represent a potential therapeutic target [12].

### 1.3. Movement across the BBB

The BBB plays an essential role in maintaining proper CNS functions by selectively regulating movement of essential nutrients and toxins into and out of the CNS. While several molecules are restricted from entering the brain, many molecules can move across the BBB membrane using four basic mechanisms, namely, passive diffusion, endocytosis, and carrier-mediated transport or active transport, depending on the characteristics of the molecule and the direction of the transport (Figure 2) [13].

Passive diffusion is the movement across a concentration gradient from high to low concentrations, without expenditure of biological energy or involvement of a carrier protein [14]. Passive diffusion can be divided into paracellular (i.e., between cells), transcellular (i.e., across cells), or via aqueous channels traversing the BBB lipid bilayer. Transcellular diffusion of compounds into the brain is dependent on their lipophilicity, as assessed by oil/water partition coefficients; the more lipophilic a compound is, the easier it could be to cross the BBB. For instance, such physicochemical characteristics have been used to facilitate CNS distribution of antidepressants and antipsychotics by inserting halogenic elements (chlorine, fluorine, bromine) in their chemical structure [15]. Transcellular diffusion is non-saturable and non-competitive [16]. Examples of drugs that can passively diffuse across biological membranes include steroids, opioids (e.g., fentanyl, methadone), benzodiazepines, and first-generation antihistamines such as diphenhydramine [14]. Small hydrophilic molecules such as nitrate, urea, glycerol, and arsenite could enter the CNS through aqueous channels [17]. Finally, due to the tight junctions connecting endothelial cells of the BBB, paracellular diffusion does not occur to any great extent under normal physiological conditions [6].

The second mechanism is endocytosis, which is defined as the vesicular transport across the BBB through receptor-mediated, adsorptive, or bulk-phase endocytosis [14,18]. Receptor-mediated endocytosis involves internalization of the substrate–receptor complex into the intracellular compartment where dissociation happens. This process can happen at both the luminal (apical) and the abluminal (basolateral) cell membranes. Transport of insulin and transferrin (iron transport protein) are classic examples of receptor-mediated endocytosis [19,20]. This transport mechanism is also energy-independent.

The third mechanism is carrier-mediated transport, through which the solute molecules move from high to low concentrations across the BBB. This membrane-bound protein-mediated transport system is an energy-independent system that could be uniporter (allows transportation of one solute at a time), symporter (carries two different solutes at the same time; the transported solute and the cotransported solute move in the same direction), or antiporter (also carries two different solutes; the transported solute and the cotransported solute move in opposite directions). This type of transporter is of utmost importance for CNS access to essential nutrients such as glucose through the glucose transporter 1, GLUT1 (SLC2A1) [14]. Centrally acting drugs may also utilize this pathway; for example, L-Dopa and gabapentin are transported through the sodium-independent large neutral amino acid transporter (LAT1 or SLC7A5) [21].

The fourth mechanism is active transport across the BBB through a carrier protein. This process is energy-dependent and often coupled with ATP hydrolysis; it enables the movement of substances against their concentration gradient [22]. There are several energy-dependent transporters expressed at the BBB endothelium that work to transport essential micronutrients, ions, and some endogenous compounds into the CNS. Active transporters could favor influx but predominantly facilitate efflux of drugs from the CNS and therefore work to regulate the entry of potentially toxic substances. This BBB crossing process is the target of many therapeutic drugs that are transported into or out of the CNS, such as antineoplastic agents, opioid analgesic drugs, HIV-1 protease inhibitors, and antibiotics [14].

While the four mechanisms are essential for maintaining brain homeostasis, the transport proteins expressed on the luminal (blood-facing) and abluminal surface of the BBB magnify the protection provided and strictly regulates the transcellular transport. Unfortunately, this process is affected by aging and age-associated disorders [23]. Both the expression level and the functionality of transport proteins go through a transformation with aging under both physiological and pathological conditions. Therefore, this review focuses on the major transporters located at the BBB and sheds light on the influence of inflammatory conditions and diseases, such as Alzheimer’s disease (AD), epilepsy, and stroke, on the expression and functionality of the BBB drug transporters, the consequential modulation of drug distribution to the brain, and its impact on drug efficacy and toxicity.

## 2. Transporters Expressed at the BBB

The gatekeepers of the BBB are the endothelial cells lining its vasculature that express a variety of transporters that strictly regulate the influx and efflux of endogenous nutrients and exogenous toxins including xenobiotics and drugs [24]. Below, an overview of the major superfamilies of transporters expressed at the BBB is provided. This includes the ATP-binding cassette (ABC) transporters and the solute carrier superfamily (SLC).

### 2.1. ATP-Binding Cassette (ABC) Transporters

The ATP-binding cassette (ABC) superfamily of transporters is perhaps the most studied family of active transporters that exists in nearly all living organisms [14,25]. The ABC family in humans includes 49 transporters grouped into seven sub-families, designated from ABCA to ABCG. Every transporter, on the basis of unique genetic and amino acid sequences, is assigned a number following the subfamily letter, e.g., ABCG2 [26].

ABC transporters are ATP-driven transmembrane proteins that function as unidirectional pumps across the BBB for various compounds including amino acids, steroids, phospholipids, ions, polysaccharides, drugs, and xenobiotic compounds [27]. At the BBB level, most ABC transporters act as efflux transporters (moving substrates outside of the CNS) and provide the BBB with a great clearance and defensive machinery [28]. Many ABC transporters are well-known mediators of a multidrug resistance (MDR) phenotype described as the simultaneous resistance to multiple structurally unrelated compounds that are not due to an independent genetic mutation that confers resistance to a single xenobiotic [14]. The most studied members of the ABC superfamily expressed at the BBB will be discussed below and include ABCB1 (P-glycoprotein, P-gp encoded by the gene *MDR1*), ABCG2 (breast cancer resistance protein, BCRP), and ABCCs (multidrug-associated resistant proteins, MRPs); they are all efflux transporters, limiting therapeutic drug entry into the brain and preventing successful response to several agents intended to treat neurological disorders. Moreover, decreased expression or function of these transporters has been described in several CNS disorders [29].

#### 2.1.1. ABCB1

ABCB1 is the classical multidrug efflux transporter that was first described in 1974 by Dr. Victor Ling at the University of Toronto through the characterization of colchicine-resistant Chinese hamster ovary cells [30,31]. Since its discovery, ABCB1 has been characterized as a 170 kDa transmembrane protein that consists of two homologous halves, each with an intracellular ATP-binding site. In the human brain, ABCB1 is extensively expressed on the luminal side of the brain endothelial cells lining the BBB. ABCB1 limit the entry of a huge variety of hydrophobic amphipathic drugs, such as cyclosporin A, digoxin, and vinblastine into the CNS by pumping them from the endothelial cells back into the blood (Figure 3) [32]. Unlike conventional transporters, which are the target of specific substrates, ABCB1 recognizes a wide variety of substrates that differ considerably in molecular size and structure [31]. This includes several antineoplastic agents, calcium channel blockers, anti-depressants, anti-epileptic agents, and several HIV-1 protease inhibitors [33].

Modulation of ABCB1 expression at the BBB has been described under various conditions [29]. These include upregulation of ABCB1 when exposed to certain drugs or hormones in vitro. In addition, a number of in vivo studies suggest that ABCB1 expression is elevated in case of epilepsy and neurodegenerative disorders such as amyotrophic lateral sclerosis (ALS) [34,35]. An increase in ABCB1 expression was reported to induce progressive pharmacoresistance to riluzole, the only FDA-approved drug thus far for ALS [36]. In contrast to ALS, downregulation of ABCB1 expression and function has been observed with aging and in patients with Alzheimer’s disease (AD) [37]. Several studies also suggested that ABCB1 function is associated with AD progression, as ABCB1 plays an important role in the clearance of amyloid-β (Aβ), the hallmark of AD [37,38]. Activity level of ABCB1 is modulated by mutations in *ABCB1*, and many recent reports have shown an association between *ABCB1* mutations and AD [39]. Some of these variants are the *3435C > T*, the *2677G > T/A* and the *1236C > T* polymorphisms [39]. While some studies have shown no association between the *3435C > T* polymorphism and AD risk, Van Assema et al. showed that the *1236C > T* polymorphism may contribute to increased Aβ deposition in the brain [39].

#### 2.1.2. ABCG2

ABCG2 was first described in 1990 by Chen et al., and was further characterized in 1998 by Doyle et al. as a polytopic plasma membrane transporter protein of 75 kDa. Doyle et al. demonstrated using the human breast carcinoma cell line MCF-7/AdrVp—which does not express the efflux transporters ABCB1 or ABCC1—the multidrug resistance aspects of ABCG2 for substrates such as the anthracycline doxorubicin (Figure 4) [40,41]. While there are at least five members of the ABCG subfamily identified in humans, ABCG2 is the primary form involved in the transport of substances across the BBB [42].

ABCG2 is expressed on the luminal side of BBB capillary endothelial cells and acts as an ATP hydrolysis-dependent efflux transporter of a wide range of structurally and chemically distinct compounds, including physiological substrates such as sulfate and glucuronide conjugates of estrone and dehydroepiandrosterone [42]. In addition, several antineoplastic agents, namely, daunorubicin, doxorubicin, imatinib, methotrexate, mitoxantrone, and topotecan are substrates of ABCG2 [42,43]. Other ABCG2 substrates from various drug classes include camptothecin derivatives, cimetidine, glyburide, sulfasalazine, prazosin, pantoprazole, methotrexate, lamivudine, nitrofurantoin, and rosuvastatin [44]. A number of photosensitizers including pheophorbide A, protoporphyrin IX, and associated compounds are also substrates of ABCG2, suggesting a potential association between ABCG2 expression and cellular resistance to photodynamic therapy [45].

#### 2.1.3. ABCCs

The ABCCs are ~190 kDa proteins whose physiological functions include organic anions and nucleotide-based analogs transport, signal transduction, and toxin secretion [14]. Although at least 12 ABCC members have been identified to data, only 9 members (ABCC1-9) are involved in the efflux transport of drugs impacting their pharmacokinetics [46]. While several ABCC members have long been associated with drug resistance across different organs of the human body, ABCC1, 2, 4, and 5 are the only isoforms with confirmed expression on the luminal side of the BBB endothelial cells. Moreover, the expression of ABCC1 and ABCC4 was also described at the basolateral side of the BBB (Figure 5) [47]. Conflicting results have been reported for the expression of ABCC3 and ABCC6 [48,49,50,51]. Collectively, the localization of the aforementioned four different ABCC isoforms at the BBB suggests that the ABCCs play an important role in regulating the efflux of drugs from the brain back to the blood and restricting the influx of several compounds into the brain parenchyma [52].

In contrast to ABCB1 and ABCG2, ABCC’s substrate structural profile is relatively restrictive [14]. Notably, the ABCC’s isoforms generally transport unconjugated anionic drugs, organic anions, and their conjugated metabolites, such as glutathione conjugates (leukotriene C4 or LTC4), glucuronides (estradiol-17-*β*-glucuronide), and glutathione disulfide (GSSG) [53]. The fact that at least nine isoforms of ABCCs show drug efflux functions with substantial overlapping substrate affinity makes it difficult to differentiate each isoforms’ specific characteristics and function [27]. This includes their potential role in the pathophysiology of several neurological diseases and response to treatment [49,54]. ABCC1 and ABCC4 exhibit a high affinity for the endogenous inflammatory mediator LTC4, suggesting a role in the transport of this compound across the BBB and a possible contribution to immune responses. In vitro studies have also shown that ABCC4 is a unique efflux transporter of prostaglandin E1 (PGE1) and PGE2 and that their transport was inhibited by nonsteroidal anti-inflammatory drugs such as indomethacin [55]. In addition, a number of antineoplastic purine nucleotide anion analogs are effluxed by ABCC4, specifically the active metabolites of 6-mercaptopurine and 6-thioguanine, and bis(pivaloyloxymethyl)-9-[2-(phosphonomethoxy)ethyl]-adenine (PMEA) [27]. ABCC4 and ABCC5 are the primary active transporters of endogenous nucleotides such as cAMP and cGMP, and may play a role in regulating their levels in the brain [56].

### 2.2. The Solute Carrier (SLCs) Superfamily

The SLC superfamily of transporters represents the largest studied superfamily of secondary active and facilitative transporters [57]. The SLC superfamily consists of almost 395 unique transporter genes divided into 65 sub-families (SLC 1-65) [58]. Among the 52 identified SLC families, the SLCO family (which includes the Organic Anion Transporting Polypeptide subfamily (OATP)), SLC1A (high-affinity glutamate transporters, EAATs), SLC2A (glucose transporters, GLUTs), SLC7A, SLC15A, SLC16A, SLC21A, SLC22A (which includes the organic anion transporters (OATs) and organic cation transporters (OCTs)), SLC28A, and SLC29A families have been identified at the BBB [59]. Unlike the ABC transporters, SLCs do not need ATP to transport substrates across biological membranes. Instead, the transport machinery is powered by an electrochemical gradient (e.g., Na^+^) or a concentration gradient, depending on the solute being transported [60]. These transporters are responsible for the transport of a wide range of substrates including organic cations and anions, peptides, monocarboxylates, steroids, signaling molecules, drugs, and drug conjugates [59].

SLC transporters are expressed on both the luminal and abluminal sides of the BBB and, like other transporters, contribute to keeping the brain protected from toxic substances while extracting essential components from the blood [61]. Although transporters in this family are capable of bidirectional transport, generally SLC transporters are considered to favor cellular uptake of drugs rather than their efflux [59]. In addition, SLC transporters can stimulate the synthesis of acetylcholine, reduce oxidative stress, act upon carnitine, and prevent neurodegeneration [62]. As such, these transporters are involved in the pathogenesis of several neurological disorders [63].

#### 2.2.1. Organic Anion Transporting Polypeptides (SLCOs and Formally OATPs)

The organic anion transporting polypeptides (SLCOs and formally OATPs) are a group of multi-specific sodium-independent membrane solute carrier transporters classified within the SLC21 family. The human SLCOs consists of 11 members, localized in several organs such as the kidney and the liver, and mediating the transport of a wide range of amphipathic substrates, including bile acids, steroid conjugates, thyroid hormones, and anionic peptides, in addition to several drugs and xenobiotics such as 3-hydroxy-3-methylglutaryl-coenzyme A (HMG-CoA) reductase inhibitors (i.e., statins), angiotensin receptor blockers, angiotensin-converting enzyme inhibitors, antibiotics, and other chemotherapeutic agents [63,64,65]. While the expression and function of the Slcos in the BBB of rodents has been well documented, the expression and function of SLCOs at the human BBB is not clearly understood [14]. Specifically, the expression of organic anion transporting polypeptide 1a4 (Slco1a4), Slco2a1, and Slco1c1 has been reported in capillary endothelial cells, capillary-enriched microvessels, and whole-brain microvessels [66]. It has also been suggested that Slco1a4, a rodent ortholog of SLCO1A2, is the primary drug transporting Slco isoform expressed at the rat BBB [67]. Recent studies using Slco1a4 (Oatp1a4) knockout mice also provide evidence for the Slco-mediated uptake of statins as knockout animals showed a reduced brain uptake of statins compared to age-matched wild-type control mice [68]. Thompson et al. showed that the function expression of Slco1a4 at the rat BBB was significantly increased after induction of inflammation and pain, which was also reflected by the increased atorvastatin brain delivery via Slco1a4 [66]. Animal studies also showed that Slco1c1 is primarily responsible for the uptake of thyroxine and conjugated sterols at the BBB, while Slco1a2 is a key player in mediating the CNS prostaglandin homeostasis [67].

In vitro data using human brain endothelial cells suggested that SLCO1A2 (OATP1A2) plays an important role in drug delivery across the BBB (including removal of organic metabolites). Expression of SLCO1A2 was the first SLCO identified in human brain endothelial cells [31,59]. Moreover, using paraffin-embedded sections of human brain endothelial cells, Gao et al. showed that SLCO2B1 (OATP2B1) is abundantly expressed at the brain capillary endothelial cells and may be involved in mediating the transport of several neuropeptides and neurosteroids across the BBB [69]. Additionally, immunohistochemical localization studies showed that SLCO1A2 is expressed on the luminal side of the BBB while Slco1A4 (rodent ortholog of human SLCO1A2) is expressed on both luminal and abluminal sides [27,70]. Additional studies are required to further understand the important role of SLCOs at the human BBB.

#### 2.2.2. Organic Anion Transporters (SLC22s Formally OATs)

The organic anion transporters (OATs)—which include SLC22A6-8 (OAT1-3), SLC22A9 (OAT7), SLC22A10 (OAT5), SLC22A11 (OAT4), the human urate transporter 1 (URAT1, also known as SLC22A12), SLC22A13 (OAT10), and SLC22A20 (OAT6)—are another important multi-specific family of transporters that belong to the SLC22 superfamily [71]. Similar to SLCOs, OATs mediate the transport of a broad range of chemically distinct endogenous and exogenous substrates such as hormones, prostaglandins, urate, dicarboxylates, and anionic neurotransmitter metabolites, as well as drugs and their associated metabolites [72].

Human SLC22A6, SLC22A7, SLC22A8, SLC22A11, SLCA12, and SLC22A13 are predominantly expressed in the human kidneys; SLC22A7, SLC22A9, and SLC22A10 are expressed in the liver; SLC22A6 and SLC22A11 are expressed in the placenta; SLC22A6, SLC22A8, SLC22A11, and SLC22A13 are expressed in the brain; and SLCA22A20 is expressed in the olfactory mucosa [70,73,74]. Proteomic analysis of human brain endothelial cells revealed differential expression of the SLC22s at the BBB level: SLC22A11 is known to be expressed at the BBB but its exact location is unclear; SLC22A6 and SLC22A8 are either not detected or have levels below the limit of quantification [60,75]. On the basis of animal studies, research found that Slc22a8 (Oat3) is expressed on the basolateral side of BBB [76]. Western blot analysis of brain capillary-enriched fraction revealed higher Slc22a8 expression with greater molecular mass compared to kidney, suggesting that its major function is to clear substrates from the brain [76]. Functional expression studies using mammalian cells showed that Slc22a8 has a broad range of substrate specificities including amphipathic organic anions, such as 17 β-glucuronide, dehydroepiandrosterone sulfate and estrone sulfate, hydrophilic organic anions such as benzylpenicillin and para-aminohippurate, and the organic cations ranitidine and cimetidine [71]. Therapeutic agents that could be transported out of the brain by SLC22A8 include several antivirals and antibiotics [59,71]. SLC22A11 is also known to be expressed at the BBB, but its exact localization is yet to be confirmed. SLC22A11 acts as a bidirectional and sodium-independent transporter for substrates such as prostaglandins, estrone sulfate, and ochratoxin A [77,78].

#### 2.2.3. Organic Cation Transporters (SLC22s Formally OCTs)

Depending on their transport capabilities, transporters that translocate organic cations are categorized into two main groups: oligospecific organic cation transporters and polyspecific organic cationic transporters [79]. Oligospecific organic cation transporters enable transport of a single main substrate (or its analogs); this group includes sodium glucose co-transporters (SLC5-family), sodium chloride co-transporters for neurotransmitters (SLC6-family), and high-affinity thiamine (vitamin B1) transporters (SLC19-family) [80,81,82,83]. In contrast, polyspecific organic cation transporters transport organic cations from a diversity of chemical structures. This includes endogenous compounds, such as dopamine, serotonin, monoamine neurotransmitters, acetylcholine and histamine, drugs and xenobiotics such as cimetidine, metformin, and phenformin, acyclovir, ganciclovir, memantine, and quinidine, and other compounds such as creatinine and guanidine [79].

Polyspecific organic cation transporters belong to a large transporter subfamily named organic cation transporters (OCTs). The transporters are part of the solute carrier family SLC22A, with three members currently described in humans: SLC22A1 (OCT1), SLC22A2 (OCT2), and SLC22A3 (OCT3); the organic cation and carnitine transporters (OCTNs) have two designated isoforms reported in humans: SLC22A4 (OCTN1) and SLC22A5 (OCTN2) [58,82]. By definition, OCTs are involved in the transport of organic cations or weakly alkaline substrates, including several orally administered drugs that contain positive charges at physiological pHs [84]. SLC22A1, SLC22A2, and SLC22A3 constitute the first SLC22 subgroup of functionally characterized transporters mostly localized to the basolateral membrane of polarized cells including the BBB endothelial cells and choroid plexus epithelial cells [14]. In vitro studies have demonstrated the expression of SLC22A1 using human brain endothelial cells that mediated the transport of the antiepileptic drug lamotrigine [85]. Other studies also linked OCTs to the transport of a number of drugs at the BBB including neurotoxin 1-methyl-4-phenyl-1,2,3,6-tetrahydropyridine (MPTP), amisulpride, oxycodone, and sulpiride [85,86,87]. Regarding the OCTNs, functional studies using in vivo and in vitro models involving the transport of acetyl-L-carnitine showed that SLC22A5 is localized at the luminal side of the BBB, while SLC22A4 has not been reported in human brain and BBB. [88,89].

## 3. Dysfunction of the BBB Transporters in Neurological Diseases

Pathological changes in the integrity and/or function of the BBB are recognized in a wide range of neurological disorders, especially those with chronic neuroinflammation, Alzheimer’s disease (AD), stroke, and epilepsy [90,91,92]. Moreover, mounting evidence supports direct involvement of functional or dysfunctional BBB transporters in the onset and progression of these diseases. Several mechanisms have been proposed including downregulation of transporter expression, their competitive or non-competitive inhibition, their induction, and genetic polymorphisms [93]. Diseases could also have profound effects on overall physiological factors, drug pharmacokinetics, drug efficacy, and toxicity. A knowledge gap remains in for our in-depth understanding of the interplay between gain/loss of function of drug transporters and their impacts on neurological diseases, and whether changes in transporter expression are a cause of consequence of these diseases. Figure 6 is a graphical abstract describing the potential impact of the abovementioned diseases on the BBB function and drug transporters.

### 3.1. Neuroinflammation and BBB

As a regulatory interface between the CNS and the immune system, the BBB is not a simple physical cellular barrier, and therefore it can both impact and can be impacted by the immune system at many levels [94]. On one hand, the BBB transports and secretes several cytokines and substances with neuroinflammatory properties. For instance, the vascular endothelial growth factor (VEGF) cytokine is a key mediator in BBB damage for several neurological diseases—VEGF is known to promote BBB leakage in the ischemic brain, brain tumors, and CNS inflammatory diseases due to a disruption of endothelial tight junction proteins [95,96,97]. On the other hand, modulation of expression and function of essential transporters (e.g., ABCB1) and even BBB disruption can be the result of acute or chronic inflammatory processes, which can have a significant impact on substance, drug, and xenobiotic transport across the BBB [94]. Neuroinflammation is a condition where the CNS is under a chronic inflammatory status characterized by continuous microglial activation usually associated with a cellular reaction to tissue injury [98,99,100]. It is basically a host defense mechanism that reestablishes the normal function and structure of the brain by putting in place mechanisms aimed at the removal of the offending agent. Unfortunately, under neuroinflammatory conditions, this defense mechanism is often associated with further CNS damage [101].

Evidence exists that acute and chronic inflammation of the BBB may affect pharmacotherapy by modulating the expression of drug transporters. For instance, Skrobik et al. evaluated the association between acute inflammation and increased risk of drug-induced coma and/or delirium in a prospective cohort study of 99 intensive unit care (ICU) patients [102]. Using midazolam and fentanyl as marker substrates of drugs regularly used in ICU patients, they demonstrated an association between coma and drug exposure that could be linked to increased BBB permeability due to increased plasma levels of the inflammatory mediators interleukin (IL)-1β and IL-6 [102]. Several preclinical studies also showed that many of the standard saturable BBB transporters are affected or impaired by neuroinflammatory events. For example, cerebral tryptophan levels increased significantly in mice injected with recombinant mouse cytokine tumor necrosis factor-alpha (TNF-α), likely due to increased permeability and transport across the BBB [94]. Moreover, enhanced insulin across the BBB was reported in mice treated with lipopolysaccharide (LPS) through modulation of the nitric oxide synthase activity [15].

One important BBB efflux transporter whose expression and function are impacted by neuroinflammation is the ABCB1/Abcb1 [103]. Bauer et al. showed that the expression of Abcb1 and its transport activity was highly upregulated in rat brain capillaries after exposure to TNF-α and endothelin-1 (ET-1), both of which are important key players in the brain’s innate immune system [104]. In contrast, Goralski et al. showed that intracranial ventricle injection of LPS in rats decreased the expression of Abcb1 in the brain, which was also reflected on increased accumulation of the Abcb1 substrate, digoxin, in brain tissue [105]. In addition to ABCB1, modulation of the expression and function of the efflux transporter ABCG2 by neuroinflammation was reported in in vitro studies using human brain endothelial cells. The expression of ABCG2 was significantly reduced by IL-1β, TNF-α, and IL-6, which was also associated by reduced uptake of the ABCG2 substrate, mitoxantrone [106]. The impact on ABCB1 was different, where IL-6 slightly reduced it expression while TNF-α increased ABCB1 expression with no impact on its activity [106]. Studies using porcine brain capillary endothelial cells found that Abcg2 was overexpressed in hydrocortisone (a corticosteroid and anti-inflammatory agent)-treated cells relative to untreated cells [107]. To further understand how ABC transporters are regulated during inflammation or infection, von Wedel-Parlow et al. analyzed the effects of TNF-α and IL-1β on multidrug resistance proteins in primary cultures of porcine brain capillary endothelial cells. The results of this study showed that TNF-α and IL-1β rapidly decreased Abcg2 mRNA expression within 6 h while expression came back to control levels after 18–24 h. Importantly, this reduction in mRNA levels within the first 6 h was counter-regulated when hydrocortisone was used [98].

In summary, although modulation of the BBB transporters as a consequence of neuroinflammation could be part of the CNS initial defense mechanism, chronic state of neuroinflammation and associated dysregulation of transporters’ function and expression could be associated with a wide range of disease-induced or drug-induced CNS damage that could lead to chronic neurodegenerative diseases such as AD, Parkinson’s disease, Huntington’s disease, or ALS [108].

### 3.2. Alzheimer’s Disease (AD)

Alzheimer’s disease (AD) is a devastating progressive neurodegenerative disorder that slowly destroys memory and thinking skills; it accounts for about 60–80% of dementia cases worldwide [109]. The classical hallmark pathologies of AD are the presence of amyloid-β (Aβ) plaques and neurofibrillary tangles (NFTs) in the brain [110]. In addition, increasing evidence supports that chronic neuroinflammation, disrupted BBB, and cerebrovascular dysfunction contribute to AD pathophysiology and cognitive dysfunction [111].

The role of the BBB breakdown and dysfunction and the dysregulation of certain BBB transporters such as the ABCB1, the lipoprotein receptor-related protein (LRP-1), and the receptor for advanced glycation end products (RAGE) have been studied extensively in vitro, in vivo, and using post-mortem brain analysis of AD patients as key factors in the development and/or the progression of the disease [112,113,114]. Recent neuroimaging studies in patients with mild cognitive impairment and early AD showed that BBB breakdown is an early event in the aging human brain that starts in the hippocampus, a critical region for memory and learning, and can lead to cognitive impairment [113]. Furthermore, the involvement of the efflux transporter ABCB1, LRP-1, and RAGE in AD originates from their well-established role in the transport and clearance of Aβ. Aβ is a proteolytic product of the amyloid precursor protein (APP). Although normal Aβ levels have been found essential in maintaining synaptic activity and neuronal survival, accumulation and diminished clearance of Aβ from the brain is neurotoxic [115]. Studies showed that the transport of Aβ across the BBB comprises a two-step process involving movement through the basolateral (brain side) and then through the apical (blood side). The LRP-1 mediates step one, which involves the incorporation of brain-derived Aβ into the endothelial cells at the basolateral membrane (i.e., transcytosis of Aβ across the brain endothelium of the BBB). Then, Aβ is believed to attach to efflux transporters such as the ABCB1, which is highly expressed on the apical membrane, and cleared into the blood. In contrary, RAGE mediates transcytosis of blood derived Aβ into the CNS (i.e., into the endothelial cells at the apical membrane) [115,116].

Therefore, downregulation of ABCB1 or LRP-1 and/or upregulation of RAGE is expected to increase Aβ brain accumulation [28]. Several in vivo preclinical studies using low concentrations of the positron emission tomography (PET) tracer [^11^C]-verapamil have initially assessed the function of Abcb1 activity at the BBB [117,118,119]. An increase in [^11^C]-verapamil concentration and binding to various sections of the brain is associated with decreased expression or function of the BBB. A recent clinical PET study compared the activity of ABCB1 at the BBB in AD patients to healthy subjects. The results of this study revealed a compromise transport activity of ABCB1 and suggested that ABCB1 downregulation might contribute to the progression of Aβ deposition in the brain; they observed a 23% increase in binding potential of [^11^C]-verapamil in the cortical region and higher binding potentials in other smaller brain regions of AD patients [38].

In another study conducted in 43 AD patients and 38 age-matched controls, Storelli et al. showed that [^11^C]-verapamil efflux and ABCB1 activity was significantly reduced in AD patients [120]. To determine whether this reduced activity was associated with reduced abundance of the transporter, they quantified abundance of ABCB1 transporter and other transporters in regions affected (hippocampus) and not affected (cerebellum) by the disease. They found that the ABCB1 abundance was decreased in the BBB of the hippocampus vs. the cerebellum in both AD and controls, and they concluded that the observed decrease was age-related rather than AD-related [120]. In contrast, the abundance of the LRP-1 protein was not significantly different in both regions for both AD and age-matched controls. Moreover, the BBB abundance of other transporters such as ABCG2, SLCO2B1, and SLC29A1 (ENT1) also decreased in the hippocampi of both groups also indicating an age-dependent decrease in density of brain microvessels [120]. The authors also concluded that other factors, such as modification of the ABCB1 transport activity (substrate affinity or transport capacity), may explain, in part, the clinical observation of reduced in vivo ABCB1 activity in AD [120]. Collectively, these results and many other reports confirmed the impact of aging and AD on BBB drug transporters activity and, more specifically, ABCB1, which was previously shown to have decreased expression and activity [121].

To directly assess the role of Abcb1 in transport of Aβ across the BBB, Cirrito et al. microinjected radiolabeled [^125^I]Aβ into the brains of Abcb1a/b^-/-^ double knockout (Abcb1-null) mice and wild-type controls [122]. The result of this study showed that ablation of Abcb1 at the BBB increased Aβ depositions. Furthermore, the authors showed a significant increase in Aβ levels in the brain interstitial fluid of amyloid precursor protein (APP)-transgenic mice within hours of treatment with a selective Abcb1 inhibitor (tariquidar) [122]. Several other in vivo studies also revealed that the chemical upregulation of Abcb1 expression at the BBB is associated with reduced Aβ brain levels and/or disposition [122,123,124].

Along with ABCB1, data suggest that ABCG2, ABCC1, ABCA1, and ABCA7 may also play a role in AD [125,126,127]. For instance, Tai et al. investigated the role of ABCG2 in addition to ABCA1 in the clearance of Aβ in human brain endothelial cells. They found that inhibition of ABCG2 and ABCB1 (by fumitremorgin C, tariquidar, and vinblastine) resulted in an increased Aβ net uptake into the cells due to a decreased efflux from the cells; they suggested that ABCG2 may act together with ABCB1 to protect the brain from influx of blood-derived Aβ [126]. Additionally, an expression analysis study of 273 BBB-related genes in the brains of AD patients vs. aged-matched nondemented controls found that the mRNA and protein levels of the BBB drug efflux transporter ABCG2 were significantly upregulated in AD brains. These results suggest that ABCG2 may act as a gatekeeper at the BBB through interactions with Aβ and prevent blood Aβ peptides from entering into the brain [128].

A genome-wide association study (GWAS) originally identified ABCA7 as a novel risk gene of AD [129]. Additional in vitro, in vivo, and human-based studies further extended this association, promoting *ABCA7* as one of the most important risk genes of AD [130,131,132]. A recent comprehensive review on *ABCA7* focusing on AD-related human genomics, transcriptomics, and methylomics concluded that human-based -omics studies give a converging proof of (partial) ABCA7 loss as a pathological mechanism in AD and indicated that future studies should make it clear if ABCA7 can be utilized as a therapeutic target for AD [132].

### 3.3. Epilepsy

Epilepsy is a chronic neurological condition associated with excessive neuronal discharges resulting in spontaneous seizures observed clinically or recorded on electroencephalograms. Around 30% of patients with epilepsy experience resistance to available therapeutic options and continue presenting seizures [133]. These often severe and frequent uncontrolled seizures significantly impact the quality of life of these patients. While aging and AD showed chronic loss of certain transporters, in epilepsy a compensatory overexpression of ABCB1 and ABCG2 transporting xenobiotics and drugs out of the CNS is a common feature and appears associated with pharmacoresistance [134,135]. Accordingly, preclinical and clinical studies have found that drug transporter expression is altered in an epileptic brain [136,137]. First, reports have suggested upregulation of ABCB1 and ABCC2 in epileptic patients [138]. Using PET scanning and R-[^11^C]-verapamil as a probe substrate for ABCB1 activity, Feldmann et al. studied the plasma-to-brain transport rate constant in a group of patients with drug-resistant mesial temporal lobe epilepsy and patients with adequate seizure control using medications [139]. Compared to patients with seizure control, pharmacoresistant patients had a lower plasma-to-brain transport rate, meaning they had a higher baseline ABCB1 activity in specific brain regions studied including ipsilateral amygdala, bilateral parahippocampus, fusiform gyrus, inferior temporal gyrus, and middle temporal gyrus. This high ABCB1 activity was also associated with higher seizure frequency in whole-brain grey matter and the hippocampus. The authors suggested that if ABCB1 contributes to pharmacoresistance, modulating and possibly inhibiting ABCB1 activity could be explored as a therapeutic strategy for drug resistant epilepsy patients. In another study, Dixit et al. measured levels of ABCG2 mRNA in patients with mesial temporal lobe epilepsy and focal cortical dysplasia. ABCG2 mRNA levels from brain resection of epileptic tissue were compared to resected sample of non-epileptic tissue from the same patient, and to brain resections from other non-epileptic patients [140]. A significant increase in ABCG2 mRNA was found in epileptic patients compared to healthy controls. Additionally, there was also higher expression of ABCG2 mRNA in epileptic tissue compared to the surrounding non-epileptic tissue from the same patient.

Mounting evidence in epilepsy suggest that a high degree of mechanistic complexity and multiple pathways drives the pathological alterations observed. The impact of epilepsy on transporters other than the ABC family also showed an abnormal expression pattern. Using immunogold electron microscopy and analysis of epileptic patients undergoing cortical resection, Cornford et al. provided evidence that the glucose transporter SLC2A1 is downregulated in regions within and surrounding the seizure focus [141]. Earlier evidence obtained from brain tissues of epileptic patients measured the distribution of the SLC transporter monocarboxylate transporter 1 (MCT1)—an important transporter of essential metabolic fuels (pyruvate, lactate, and ketone bodies), certain amino acids, and possibly with a role in transporting acidic drugs such as valproic acid across the BBB—showed that MCT1 is severely deficient on BBB microvessels of patients with drug-resistant temporal lobe epilepsy [142]. This finding may explain the reduced brain energy stores, loss of the GABAergic inhibition, and lowered threshold for recurrent seizures [142].

Antiepileptic drugs such as carbamazepine are substrates of drug transporters such as ABCB1 and ABCG2. Although upregulation of these transporters has been demonstrated in clinical studies before, it is not known whether this leads to reduced anti-seizure drug concentrations in interstitial fluid. Rambeck et al. attempted to determine concentrations of antiepileptic drugs in serum, subarachnoid cerebrospinal fluid (CSF), and extracellular space (ECS) of epileptic tissues [143]. Specific drug concentrations in the ECS were measured using intraoperative microdialysis catheters. ECS concentrations were found to be highly variable between patients (compared to relatively low variability in serum concentrations), and ECS concentrations of many anti-seizure medications were lower than those measured in CSF in pharmacoresistant patients. However, the authors noted that without data from nonepileptic patients, it is difficult to conclude whether this is exclusively due to increased transporter expression in the epileptic tissue [143].

Following the notion that increased expression of transporters may result in decreased distribution of anti-seizure medications to the brain, it is logical to propose that inhibiting drug transporters can help increase brain accumulation of anti-epileptic drugs. Accordingly, Dickens et al. used the ABCB1 inhibitor tariquidar to determine whether it could modulate the transport of phenytoin, levetiracetam, and/or carbamazepine in seven different cell systems. Results obtained in all seven cell lines demonstrated that neither lamotrigine nor carbamazepine are substrates of ABCB1. Phenytoin showed characteristics of a weak substrate in polarized kidney cells (MDCK and LLC-PK1); however, it was not a substrate in any other cell models [144]. Subsequent preclinical studies conducted in a rat model have shown that tariquidar co-administration not only increased phenytoin levels in the ventral hippocampus and entorhinal cortex, but also improved seizure control in these animals [136]. Luna-Tortos et al. showed that another anti-seizure drug, topiramate, may be an ABCB1 substrate using polarized kidney cells (MDCK and LLC) [145].

Although these preclinical data were encouraging, similar results have not been reported in patients. An attempt to relate ABCB1 activity to antiepileptic drugs’ efficacy has been made using pharmacogenomic studies. It was initially reported that the *CC* genotype at *ABCB1 3435*, (impaired function) may be associated with drug resistant epilepsy [146]. Sills et al. conducted a similar study comparing variants at the *ABCB1 3435* position with anti-seizure drug response. These authors did not find a significant relation between the *CC* genotype and drug non-responders (drug resistant epilepsy) [147]. Thus, though it is clear that epilepsy may alter the expression of drug transporters, it is unclear whether this plays a role in modulating therapeutic efficacy of anti-epileptic medication [148,149].

The impact of drug transporter-induced pharmacoresistance in epileptic patients could not be only limited to antiepileptic drugs. For instance, in patients with brain tumors (e.g., glioma, glioblastoma), the presence of epilepsy is considered the most significant risk factor for chronic disability [150,151]. These patients experience a very complex therapeutic profile including chemoresistance for drugs such as temozolomide [152]. They also present pharmacokinetic drug–drug interactions and increased incidents of seizures or refractory epilepsy due to pharmacoresistance requiring close antiepileptic drug monitoring and a multidisciplinary therapeutic approach with proper selection of chemotherapy and antiepileptic drugs [150,151,153].

### 3.4. Stroke

Globally, stroke is the second leading cause of death and a major cause of serious long-term disability [154,155]. About 87% of all strokes are ischemic strokes characterized by impaired blood supply to an affected brain region [156]. A hallmark of ischemic stroke is the dysfunction of the BBB. Preclinical and clinical studies have demonstrated that ischemic stroke disrupts the integrity of the BBB through breakdown of the tight junction protein complexes, which could lead to vasogenic edema and hemorrhagic transformation [90,157,158]. Several studies also reported a dysregulation of the BBB transporter’s expression after ischemic stroke or hypoxia/reperfusion [90,159,160]. A number of preclinical studies reported an upregulation in the expression of Abcb1, Abcg2, Slc21a5, and the glucose transporters Glut1 and Glut3 [160,161,162].

Further studies described below have tried to correlate upregulation of transporters to drug delivery into the ischemic brain and suggested a potential signaling pathway that induced the upregulation [160,163]. For example, in vitro studies using human derived cell lines suggested that hypoxia is associated with upregulation of the *ABCB1 (MDR1)* gene expression and increased ABCB1 protein expression through activation of hypoxia-inducible factor alpha (HIF-1α) and nuclear factor κB [160,163]. Spudich et al. confirmed an upregulation of Abcb1 after focal ischemia in rats and suggested that inhibition of Abcb1 may greatly facilitate neuroprotective treatments [164]. In their study, they evaluated the permeability of tacrolimus and rifampicin, two known neuroprotectants and substrates of Abcb1 before and after Abcb1 inhibition using tariquidar [164]. They found increased accumulation of both agents (14- and 4-fold for tacrolimus and rifampicin, respectively) in the ischemic brain compared to only twofold accumulation in the contralateral non-ischemic brain. This finding led the authors to suggest that there is a possibility to improve delivery of neuroprotective Abcb1 substrates to the brain by using Abcb1 inhibitors [164].

Another family of transporters that may have utility in delivering neuroprotective drugs across the BBB is the SLCO family. In rodent brain, Slco1a4 is the primary drug transporting Slco isoform expressed at the BBB [67]. In rat brain microvessels, cerebral hypoxia and subsequent reoxygenation stress increased Slco1a4 expression [66]. Thompson et al. examined regulation and functional expression of Slco1a4 and the potential neuroprotective effect of statins (Oat1a4 substrates) [66]. Results of this study showed that the pharmacological inhibition of transforming growth factor-β (TGF-β)/activin receptor-like kinase 5 (ALK5) signaling increased Oatp1a4 functional expression and that targeting Oatp1a4 could be a viable neuroprotective strategy to increase statin levels in the brain and reduce stroke volume [66]. The suggested neuroprotective mechanisms of statins would involve their antioxidant properties, leading to upregulation of endothelial nitric oxide synthase, inhibition of inducible nitric oxide synthase, and attenuation of the inflammatory cytokine responses during ischemia [165,166,167]. However, achieving therapeutic levels of drugs such as statins in the brain is a challenge due to low BBB permeability although upregulated Slco1a4 after hypoxia-reperfusion injury may help target statins to the brain [166]. Additionally, the authors suggest that modulating the expression of Slco1a4 by targeting the TGF- β/ALK5 pathway may be a potential strategy to increase statin levels in the brain and reduce stroke volume [66]. However, this preclinical evidence has not been reproduced in clinical studies yet.

### 3.5. Amyotrophic Lateral Sclerosis (ALS)

Amyotrophic lateral sclerosis (ALS) also known as Lou Gehrig’s disease, is a progressive neurodegenerative paralytic disease characterized by degradation and gliosis of the upper and lower motor neurons in the brain and the spinal cord, respectively [168]. The etiology of ALS is a multi-faceted, elusive, and complex disease with a worldwide incidence of approximately 1.6 to 2 cases per 100,000 persons each year [169,170]. While most cases of ALS are sporadic, almost 5–10% of cases are linked to familial patterns of mutations [171,172,173,174,175,176,177,178,179]. The complexity of ALS etiology and pathological changes including the disruption in the BBB integrity and function constraint the development of a successful and curative ALS treatment [35].

In the mid-1980s, clinical studies reported that ALS patients had an altered BBB integrity and permeability [180,181,182]. This finding was supported by abnormal levels of serum proteins and complement in CSF and the detection of blood-borne substances in the CNS tissues [180,181,182]. More recent studies in animals and humans described the association between BBB disruption and ALS disease [179,183,184]. In addition to the damaged BBB integrity, evidence suggest that the expression and activity of BBB drug transporters are altered in ALS patients [29,185,186]. Indeed, modulation in the expression and function of ABCB1 and ABCG2 were extensively described in literature [29,185,186,187]. Several studies using different SOD1 transgenic mouse and rat models of ALS clearly showed a significant increase in both expression and function of Abcb1 [188,189,190]. In addition, these studies highlighted the role of Abcb1 in mediating pharmacoresistance in the SOD1 mouse models of ALS [188,189,191]. However, inconsistencies were observed regarding the expression and function of ABCG2 in ALS. For instance, Milane et al. investigated the activity and expression of Abcb1 and Abcg2 in brain microvessels of an ALS transgenic mouse model [188]. Their results showed a 1.5-fold increase in Abcb1 expression and no change in Abcg2 expression in ALS transgenic mice vs. the wild-type group (control) [188]. In opposite, Jablonski et al. reported a selective increase in expression of Abcg2 and Abcb1 using two ALS mouse models [185]. This finding was also confirmed in spinal cord extracts of ALS patients [185].

Riluzole (RLZ), one of the two FDA-approved drugs for ALS management, is a substrate of ABCB1 and ABCG2 [192,193]. A study showed a significant increase in RLZ CNS penetration and significant improvements in behavioral and efficacy measures when RLZ was administered concomitantly with elacridar (a potent ABCB1 and ABCG2 inhibitor) [36]. A recent study by Yang et al. assessed a liposomal co-delivery system that could facilitate the penetration of RLZ in brain cells by reducing its efflux using verapamil as a ABCB1 inhibitor [192]. Their results demonstrated that under ALS-mimic conditions in mouse brain endothelial bEND.3 cells, the treatment with the cocktail verapamil liposomes inhibited Abcb1 [192]. Their study also demonstrated that endothelial brain cells exposed to the cocktail verapamil-rhodamine 123 liposomes restored the uptake of rhodamine 123 (Abcb1 substrate). Similarly, the cocktail verapamil treatment significantly improved the RLZ uptake in an in vitro BBB cell model [192]. Further mechanistic studies are still needed to clearly understand the potential of targeting ABCB1 and/or ABCG2 for developing and improving ALS therapies [29].

### 3.6. Multiple Sclerosis

Multiple sclerosis (MS) is a chronic inflammatory autoimmune CNS disease that cause demyelination, axonal loss, and neurodegeneration [194]. MS is estimated to affect 2.5 million people worldwide and nearly 1 million people in the United States [195,196]. The pathophysiology of MS is complex and is mainly related to two fundamental process described as (1) focal chronic inflammation inducing macroscopic plaques and BBB injury, and (2) neurodegeneration with microscopic injury in the neurons, axons, and synapses [197]. Although the cause of MS is still unknown, it is believed that MS is triggered by a combination of factors such as environmental, genetic, and non-genetic factors [198].

Similar to chronic neuroinflammation, MS compromises the integrity and function of the BBB. Evidence suggests that dysfunctional transporters at the BBB have been implicated in MS; however, the exact mechanism is uncertain. Several studies identified an important role for ABCB1, ABCG2, ABCC1, and ABCC2 in neuroinflammatory processes underlying MS pathology [199,200,201,202]. Kooij et al. reported a reduced vascular expression of ABCB1 in active MS lesions from MS human brain tissue [199]. Their study also showed that, in vivo, the expression and function of Abcb1 at the BBB was significantly impaired in MS animal model [200]. The same group investigated the expression pattern of different ABC transporters (ABCB1, ABCG2, ABCC1, and ABCC2) in human MS lesions [199]. ABCB1 cerebrovascular expression was reduced in active and chronic inactive MS lesions [199]. In contrast, reactive primary astrocytes showed an increase in both ABCB1 and ABCC1 expression in human brain tissues of MS patients compared to control [199]. They proposed a potential new pathophysiological role for ABCB1 and ABCC1 on reactive astrocytes—they could contribute to the inflammatory process by mediating immune cell migration and aggravating inflammatory attack during MS pathogenesis [199].

## 4. Conclusions

As the body’s central processing unit, the brain is equipped with a very efficient and complex firewall system named the BBB. In this review, we discussed the critical role of the BBB with a major focus on transporter proteins and their central physiological function in preserving cerebral homeostasis. The interplay between physiology and pathophysiology of transporters in the human brain has gained much interest in recent years. Although the role of the BBB drug transporters in regulating distribution of therapeutics in the CNS has been known for several decades, it is becoming increasing clear that BBB transporters are involved in the onset and progression of several neurological diseases (major findings are summarized in Table 1). Still, we remain far from having a comprehensive understanding of the specific and dynamic roles of each transporter and their direct association with various neurological diseases. In addition, the large body of evidence regarding the impact of BBB drug transporters as potential targets for a wide array of neurological diseases is based on preclinical in vitro and in vivo studies with a limited number of clinical findings. Nevertheless, several recent clinical efforts utilizing current and emerging advanced analytical methods (e.g., proteomics) coupled with diagnostic neuroimaging studies (e.g., PET scans, magnetic resonance imaging (MRI)) have significantly improved our understanding and appreciation of the role of specific BBB transporters in certain neurological diseases such as AD where specific transporters such as ABCB1 play a role in the disease pathology. The impact of other neurological diseases such as epilepsy on the expression and function of the BBB drug transporters ABCB1 and ABCG2, as well as other efflux or influx transporters, may be different.

Recent research and neuroimaging studies greatly advance our understanding of BBB drug transporters, making them potential targets for several neurological diseases. Our awareness has also heightened about the impact of BBB dysfunction in the pathology of a wide range of CNS diseases; this provides invaluable insights into targeting BBB transporters as a way to prevent or slow the progression of certain neurological diseases. At the same time, it remains challenging to understand the complex physiological functions of these transporters. From many examples discussed in this review, various neurological diseases can have two completely different effects on the activity and expression of BBB transporters. Therefore, a better understanding of molecular and cellular mechanisms underlying function and expression and of these transporters will significantly improve the way we think about brain diseases and could lead to the discovery of novel BBB targets and approaches for therapeutic solutions.

## Figures and Tables

**Figure 1 ijms-22-03742-f001:**
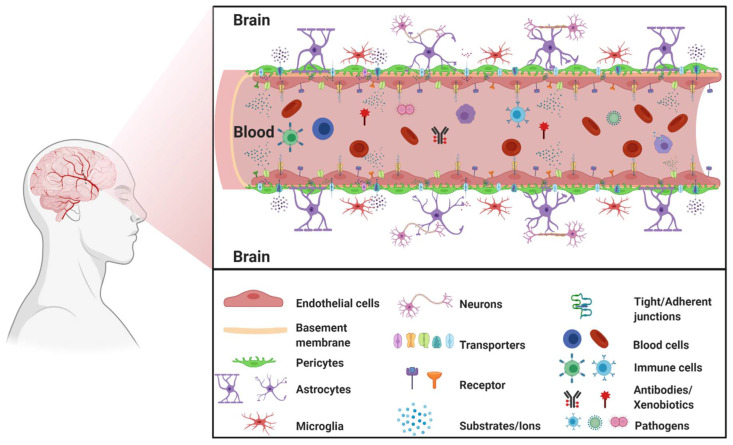
The blood–brain barrier. (Created with BioRender, accessed on 27 January 2021).

**Figure 2 ijms-22-03742-f002:**
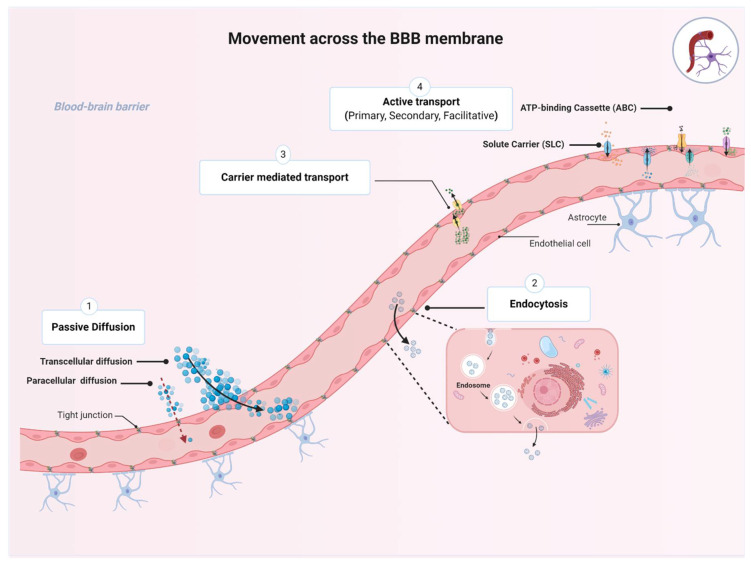
Transport pathways across the blood–brain barrier (BBB). The passage of various molecules through the brain implies four basic mechanisms allowing specific molecules to move across the BBB membrane including: (**1**) the passive diffusion (spontaneous movement across a concentration gradient), (**2**) endocytosis (receptor-mediated, adsorptive, or bulk-phase endocytosis), (**3**) carrier-mediated transport (movement across a concentration gradient and energy independent), and (**4**) active transport (movement of molecules against a concentration gradient and energy dependent). Collectively, the four mechanism plays an essential role for maintaining brain homeostasis. (Created with BioRender, accessed on 27 January 2021).

**Figure 3 ijms-22-03742-f003:**
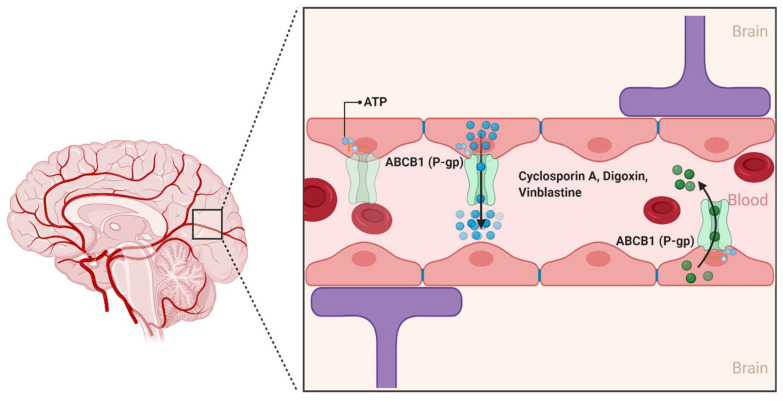
The ABCB1 transporter at the blood–brain interfaces. In the human brain, the ABCB1 transporter is found on the luminal membrane side and restricts the penetration of compounds in the brain by limiting the uptake of several substrate drugs. (Created with BioRender, accessed on 27 January 2021).

**Figure 4 ijms-22-03742-f004:**
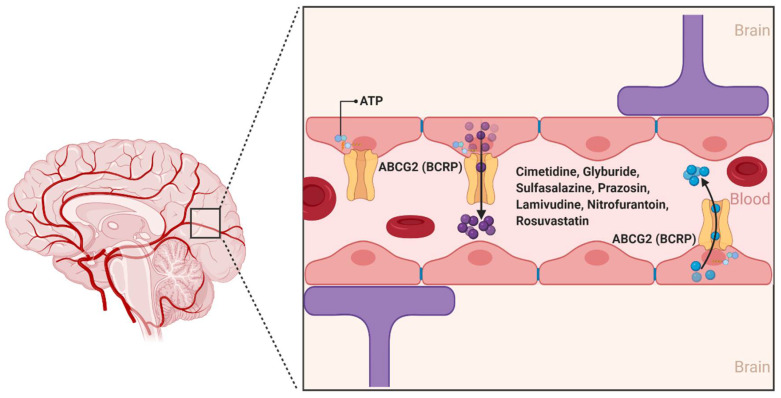
The ABCG2 transporter at the blood–brain interfaces. ABCG2 was originally identified as the breast cancer resistance protein (BCRP). In the human brain, ABCG2 is expressed on the luminal membrane side and plays an important role in limiting exposure of drug molecules to the CNS. (Created with BioRender, accessed on 27 January 2021).

**Figure 5 ijms-22-03742-f005:**
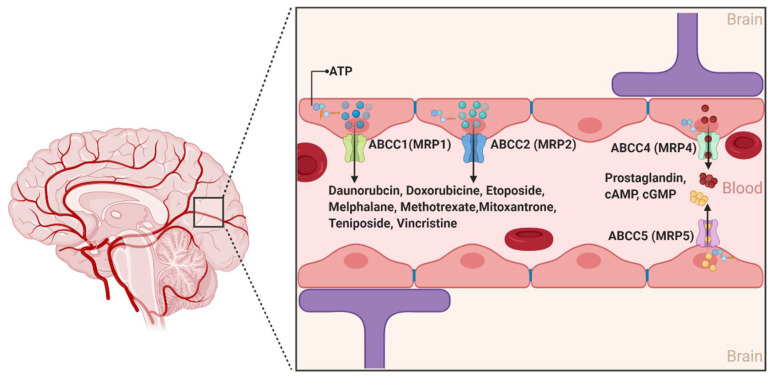
The ABCC transporters at the blood–brain interfaces. In the human brain, ABCC transporters are preferentially expressed on the luminal sides. ABCC proteins are active efflux transporters playing a role in the extrusion of several molecules. (Created with BioRender, accessed on 27 January 2021).

**Figure 6 ijms-22-03742-f006:**
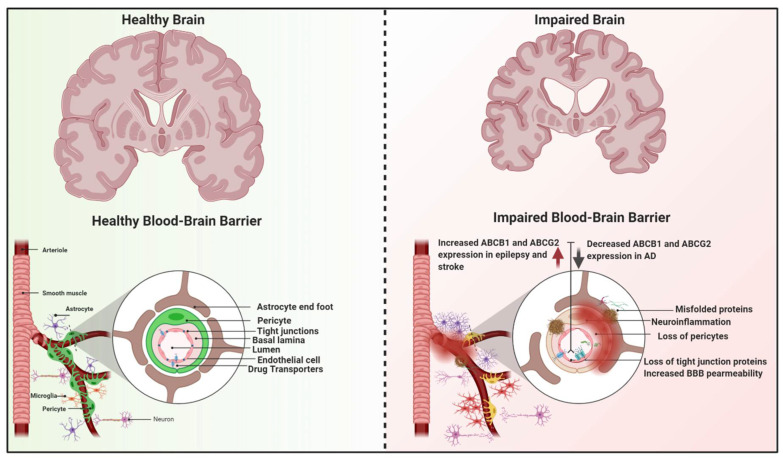
The blood–brain barrier (BBB) under healthy and diseased conditions. The BBB is formed by endothelial cells of the brain capillaries that are connected with tight junctions and supported by the neurovascular unit. Under neurological diseases and in aging, the BBB may undergo modulations affecting its permeability and integrity. Failure in BBB function may lead to a state of brain dyshomeostasis, neurodegeneration and brain atrophy. (Created with BioRender, accessed on 27 January 2021).

**Table 1 ijms-22-03742-t001:** Summary on the influence of inflammatory conditions and diseases on the expression and functionality of the BBB drug transporters.

Transporter	BBB Expression	Function	Disease Impacted and Key Findings	References
**ATP-binding cassette (ABC) transporters**				
ABCB1 (P-gp)	Luminal (apical)	Efflux	**Neuroinflammation**: ABCB1 expression and/or function dysregulation were described in states of chronic neuroinflammation. **AD**: ABCB1 plays a role in the clearance of Aβ peptides from the brain across the BBB. ABCB1 decreased expression, and dysregulation might contribute to the progression of Aβ deposition in the brain. **Epilepsy:** ABCB1 overexpression and increased activity suggested possible explanation for the lack of drug response and reduced brain accumulation of anti-epileptic drugs. **Stroke:** ABCB1 expression and activity were mostly upregulated in ischemia and stroke. ABCB1 inhibitors were proposed to improve therapeutic effects of neuroprotective drugs in ischemic brain. **ALS:** ABCB1 overexpression and increased function were extensively reported and highlighted the role of ABCB1 in mediating pharmacoresistance in ALS. **MS:** ABCB1 cerebrovascular expression was reduced and function was impaired. ABCB1 expression was increased in reactive astrocytes in human brain of MS patients.	[122,185,188,199,200,203,204,205,206,207,208,209,210,211,212,213,214,215,216]
ABCG2 (BCRP)	Luminal (apical)	Efflux	**Neuroinflammation**: ABCG2 expression and/or function dysregulation were reported in chronic neuroinflammatory conditions. **AD**: ABCG2 expression and function were upregulated in AD. ABCG2 was shown to participate in the clearance of Aβ peptides and transport across the BBB and, possibly, acting as a gatekeeper at the BBB (by interacting and preventing the entry of blood Aβ peptides in the brain). **Epilepsy:** ABCG2 expression and activity were upregulated in epilepsy. ABCG2 could decrease brain accumulation of anti-epileptic drugs. **Stroke:** ABCG2 expression and activity were mostly increased in ischemia and stroke. **ALS:** ABCG2 expression and function were mainly upregulated in ALS and played a role in pharmacoresistance in ALS.	[128,199,211,216,217,218,219]
ABCC1 (MRP1)	Luminal (apical) and basolateral	Efflux	**AD**: ABCC1 has an important role in cerebral Aβ protein clearance across the BBB. Decreased expression and dysregulation might contribute to the progression of Aβ deposition and accumulation in the brain. **Epilepsy**: ABCC1 overexpression and increased activity induced by status epilepticus were associated with reduced anti-epileptic drugs levels in the brain and reduced efficacy. **Stroke**: ABCC1 dysregulation in response to ischemia was reported. Studies indicated a reduced expression of ABCC1 in ischemic brain capillaries. **MS**: ABCC1 expression was increased in reactive astrocytes in the brain of MS patients.	[127,199,220,221,222,223]
ABCC2 (MRP2)	Luminal (apical)	Efflux	**Neuroinflammation**: Abcc2 transport activity and protein expression were increased in rat or mouse brain capillaries in response to activation of Nrf2. **Epilepsy**: ABCC2 overexpression and increased activity induced by status epilepticus was associated with reduced anti-epileptic drugs levels in the brain and reduced efficacy.	[137,199,222]
ABCC4 (MRP4)	Luminal (apical) and basolateral	Efflux	**AD:** ABCC4 increased protein expression was detected at in the hippocampal sections from AD brain.	[207,222]
ABCC5 (MRP5)	Luminal (apical)	Efflux	**Stroke:** Abcc5 expression in rat brains was upregulated following ischemic events induced by middle cerebral artery occlusion.	[137,216]
ABCA7	Not clear	Efflux	**AD:** GWAS studies identified risk variants in *ABCA7* with direct functional impact suggesting evidence of its role in AD risk. Its role could be linked to the regulation of cholesterol homeostasis and Aβ efflux at the BBB. Significant increase of insoluble Aβ and plaque load in the brains of Abca7-deficient mice. The underlying mechanism of ABCA7 role in AD pathogenesis is still unclear.	[132,224,225,226,227,228]
**Organic Anion Transporting Polypeptides**				
SLCO1A2 (OATP1A2)	Luminal (apical)	Uptake	**Stroke**: The rodent ortholog of human SLCO1A2, Slco1a4, expression was increased following cerebral hypoxia and subsequent reoxygenation stress, a central component of ischemic stroke.	[31,66,70]
SLCO2B1 (OATP2B1)	Luminal (apical)	Uptake	**Aging:** SLCO2B1 expression was decreased in the gray matter BBB of both AD and age-matched controls in different brain regions.	[120]
**Others**				
SLC27A1 (FATP1)	Basolateral	Efflux	**AD:** SLC27A1 expression was shown to be reduced and impaired after exposure to Aβ peptides in human brain endothelial cells. SLC27A1 was shown to impair the transport of docosahexaenoic acid into the brain.	[229,230]
SLC29A1 (ENT1)	Not clear	Efflux/uptake?	**Aging:** SLC29A1 expression was decreased in the gray matter BBB of both AD and age-matched controls in different brain regions.	[120]
SLC2A1 (GLUT1)	Luminal (apical) and basolateral	Uptake	**Epilepsy**: SLC2A1 was downregulated in regions within and surrounding the seizure focus.	[141]

Abbreviation: Aβ, amyloid beta; AD, Alzheimer’s disease; ALS, amyotrophic lateral sclerosis; BBB, blood–brain barrier; BCRP, breast cancer resistance protein; ENT1, equilibrative nucleoside transporter 1; FATP1, fatty acid transport protein 1; GLUT1, glucose transporter 1; GWAS, genome-wide association studies; MRP, multidrug resistance-associated protein; MS, multiple sclerosis; Nrf2: nuclear factor erythroid-derived 2-like factor; OATP, organic anion transporting polypeptides.

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
