# Peer review of "Disease-Induced Modulation of Drug Transporters at the Blood–Brain Barrier Level"

_ijms, 2021, doi:10.3390/ijms22073742_

Round 1

Reviewer 1 Report

The authors have presented a very well written review highlighting a choice selection of transporters located at the blood-brain barrier, and detailing their specific role in preserving CNS homeostasis. Towards that, they have very nicely outlined the state-of-play in our current understanding of blood-brain barrier transport mechanisms, before narrowing the focus on a number that have a crucial role in preserving cerebral homeostasis and discussing their reflected efficacy in the face of age and other debilitating environments such as inflammation etc. Taken together, this is a timely piece which reflects on, and highlights the importance, of research in this area – especially as the average age in society is ever increasing, and a great number of neuro-associated conditions are increasing in incidence as a result.

In reviewing the manuscript I had a small number of points which need attention. The following should be revised when preparing a suitable revision.

  1. The figure legends could be developed more/reviewed. For example, Figure 1 has a great deal of content which I believe should actually be part of the opening introduction, but possibly has merged to become the figure legend. In contrast, the remaining figure legends are very scant on detail, with little to no descriptive power in the words to accompany what are complex, and very nicely illustrated diagrams. The authors must review their approach to figure legends in advance of any resubmission.
  2. There is a lot of really nice details included in this review, and I would wonder would it be worth including a table which summarises the findings, and the sources of the information. This might also give the authors an opportunity to include other transporters, without having to devote paragraphs or headings towards them. The authors should strongly consider including a table in any resubmission.
  3. While written very well, there are a number of typos within the review, in particular, the loss of capitalisation when discussing certain transporters. This is a minor critique, but one that appears quite often throughout. The authors should revise the manuscript and reduce the instances of this.   

Author Response

The authors have presented a very well written review highlighting a choice selection of transporters located at the blood-brain barrier and detailing their specific role in preserving CNS homeostasis. Towards that, they have very nicely outlined the state-of-play in our current understanding of blood-brain barrier transport mechanisms, before narrowing the focus on a number that have a crucial role in preserving cerebral homeostasis and discussing their reflected efficacy in the face of age and other debilitating environments such as inflammation etc. Taken together, this is a timely piece which reflects on, and highlights the importance, of research in this area – especially as the average age in society is ever increasing, and a great number of neuro-associated conditions are increasing in incidence as a result.

In reviewing the manuscript, I had a small number of points which need attention. The following should be revised when preparing a suitable revision.

  1. The figure legends could be developed more/reviewed. For example, Figure 1 has a great deal of content which I believe should actually be part of the opening introduction, but possibly has merged to become the figure legend. In contrast, the remaining figure legends are very scant on detail, with little to no descriptive power in the words to accompany what are complex, and very nicely illustrated diagrams. The authors must review their approach to figure legends in advance of any resubmission.

Response: We thank the reviewer for his comment, the legends for figures 2-6 were modified to include more details to describe each figure as follows:

Figure 2. Transport pathways across the blood-brain barrier. The passage of various molecules through the brain implies four basic mechanisms allowing specific molecules to move across the BBB membrane including : 1) the passive diffusion (spontaneous movement across a concentration gradient), 2) endocytosis (receptor-mediated, adsorptive or bulk-phase endocytosis), 3) carrier-mediated transport (movement across a concentration gradient and energy independent), and 4) active transport (movement of molecules against a concentration gradient and energy dependent). Collectively, the four mechanism plays an essential role for maintaining brain homeostasis. (Created with BioRender).

Figure 3. The ABCB1 transporter at the blood-brain interfaces. In the human brain, the ABCB1 is found on the luminal membrane side and restricts brain penetration by limiting the uptake of several substrate drugs. (Created with BioRender).

Figure 4. The ABCG2 transporter at the blood-brain interfaces. ABCG2 was originally identified as the breast cancer resistance protein (BCRP). In human brain, the ABCG2 is expressed on the luminal membrane side and plays an important role in limiting exposure of drug molecules to the CNS. (Created with BioRender).

Figure 5. The ABCC transporters at the blood-brain interfaces. In the human brain, ABCCs transporters are preferentially expressed on the luminal sides. The ABCCs transporters are active efflux transporters playing a role in the extrusion of several molecules.  (Created with BioRender).

Figure 6. Graphical abstract of the blood-brain barrier (BBB) under healthy and diseased conditions. The BBB is formed by endothelial cells of the brain capillaries that are connected with tight junctions and supported by the neurovascular unit. Under neurological diseases and in aging, the BBB may undergo modulations affecting its permeability and integrity. Failure of BBB function may lead to a state of brain dyshomeostasis, neurodegeneration and brain atrophy. (Created with BioRender).

  1. There is a lot of really nice details included in this review, and I would wonder would it be worth including a table which summarizes the findings, and the sources of the information. This might also give the authors an opportunity to include other transporters, without having to devote paragraphs or headings towards them. The authors should strongly consider including a table in any resubmission.

Response: As suggested by the reviewer, a table that summarizes findings highlighted in the manuscript is now provided.

  1. While written very well, there are a number of typos within the review, in particular, the loss of capitalisation when discussing certain transporters. This is a minor critique, but one that appears quite often throughout. The authors should revise the manuscript and reduce the instances of this.

Response: The manuscript has been revised in order to correct any typos. In human (human specific or human homolog), symbols (gene or protein) are written all caps. However, symbols for rodents begin with an upper-case letter (only the first letter) followed by all lowercase letter in lowercase. When transporters refer to a rodent (mouse or rat studies), their corresponding nomenclature is used.

Reviewer 2 Report

The manuscript provides a good review of modulation of transporters at the BBB under neurological diseases. The review covers the roles of major transporters in inflammation, AD, epilepsy, and stroke, respectively, which are interesting and highlighting some of studies published in the most recent years.  I recommend the publication of this manuscript with the following concerns addressed.

Section 3 discussed AD, epilepsy and stroke, but if the authors could include diseases like amyotrophic lateral sclerosis (ALS) and multiple sclerosis in Section 3, it would provide a more comprehensive overview.  As the authors also mentioned in section 2.1.1 that  “ABCB1 expression elevated in case of … (ALS)”.

Among the cytokines, vascular endothelial growth factor  (VEGF),  which  is  upregulated  in  many  brain diseases, is known  to  promote  BBB  leakage  in  the  ischemic  brain, in brain tumors, and  in  CNS  inflammatory diseases  by  disrupting  endothelial tight  junction  proteins. By using submicrometer-resolution multiphoton fluorescence microscopy with a  longer  penetration  depth  into  brain parenchyma, this paper first sought to quantify temporal VEGF effects on BBB permeability to various-sized molecules: Shi, L., Zeng, M. and Fu, B.M. (2014), Temporal effects of vascular endothelial growth factor and 3,5‐cyclic monophosphate on blood–brain barrier solute permeability in vivo. Journal of Neuroscience Research, 92: 1678-1689. https://doi.org/10.1002/jnr.23457

The authors used transporter acronyms differently all through the manuscript, such as ABCB1 & Abcb1, ABCG2  & Abcg2, SLCOs & Slcos, these better be consistent.

Line 185, “ABCG2” – this is Section 2.1.2, therefore,  Section “2.1.2 ABCCs” (Line 206) is Section “2.1.3”.

Author Response

Reviewer 2

 The manuscript provides a good review of modulation of transporters at the BBB under neurological diseases. The review covers the roles of major transporters in inflammation, AD, epilepsy, and stroke, respectively, which are interesting and highlighting some of studies published in the most recent years.  I recommend the publication of this manuscript with the following concerns addressed.

  1. Section 3 discussed AD, epilepsy and stroke, but if the authors could include diseases like amyotrophic lateral sclerosis (ALS) and multiple sclerosis in Section 3, it would provide a more comprehensive overview. As the authors also mentioned in section 2.1.1 that  “ABCB1 expression elevated in case of … (ALS)”.

Response: We thank the reviewer for this constructive comment. Accordingly, we added the following two sub-sections:  3.5. “Amyotrophic lateral sclerosis (ALS) and 3.6. Multiple Sclerosis in Section 3.

3.5 Amyotrophic lateral sclerosis (ALS)

Amyotrophic lateral sclerosis (ALS) also known as Lou Gehrig's disease, is a progressive neurodegenerative paralytic disease characterized by degradation and gliosis of the upper and lower motor neurons in the brain and the spinal cord, respectively.[166]. The etiology of ALS is a multi-faceted, elusive, and complex disease with a worldwide incidence of approximately 1.6 to 2 cases per 100,000 persons each year.[167,168] While most cases of ALS are sporadic, almost 5-10% of cases are linked to familial patterns of mutations.[169-177] The complexity of ALS etiology and pathological changes including the disruption in the BBB integrity and function constraint the development of a successful and curative ALS treatment.[35]

In the mid-80s, clinical studies reported that ALS patients had an altered BBB integrity and permeability.[178-180] This finding was supported by abnormal levels of serum proteins and complement in CSF and the detection of blood-borne substances in the CNS tissues.[178-180] More recent studies in animals and humans described the association between BBB disruption and ALS disease.[177,181,182] In addition to the damaged BBB integrity, evidence suggest that the expression and activity of BBB drug transporters are altered in ALS patients.[29,183,184] Indeed, modulation in the expression and function of ABCB1 and ABCG2 were extensively described in literature.[29,183-185] Several studies using different SOD1 transgenic mouse and rat models of ALS clearly showed a significant increase in both expression and function of Abcb1.[186-188] In addition, these studies highlighted the role of Abcb1 in mediating pharmacoresistance in the SOD1 mouse models of ALS.[186,187,189] However, inconsistencies were observed regarding the expression and function of ABCG2 in ALS. For instance, Milane et al. investigated the activity and expression of Abcb1 and Abcg2 in brain microvessels of an ALS transgenic mouse model.[186] Their results showed a 1.5-fold increase in Abcb1 expression and no change in Abcg2 expression in ALS transgenic mice vs the wild-type group (control).[186] In opposite, Jablonski et al. reported a selective increase in expression of Abcg2 and Abcb1 using two ALS mouse models.[183] This finding was also confirmed in spinal cord extracts of ALS patients.[183]

Riluzole (RLZ), one of the 2 FDA approved drug for ALS management, is a substrate of ABCB1 and ABCG2.[190,191] A study showed a significant increase in RLZ CNS penetration and significant improvements in behavioral and efficacy measures when RLZ was administered concomitantly with elacridar (a potent ABCB1 and ABCG2 inhibitor).[36] A recent study by Yang et al. assessed a liposomal co-delivery system that could facilitate the penetration of RLZ in brain cells by reducing its efflux using verapamil as a ABCB1 inhibitor.[190] Their results demonstrated that under ALS-mimic conditions in mouse brain endothelial bEND.3 cells, the treatment with the cocktail verapamil liposomes inhibited Abcb1.[190] Their study also demonstrated that endothelial brain cells exposed to the cocktail verapamil-rhodamine 123 liposomes restored the uptake of rhodamine 123 (Abcb1 substrate). Similarly, the cocktail verapamil treatment significantly improved the RLZ uptake in an in vitro BBB cell model.[190] Further mechanistic studies are still needed to clearly understand the potential of targeting ABCB1 and/or ABCG2 for developing and improving ALS therapies.[29]

3.6. Multiple Sclerosis 

Multiple sclerosis (MS) is a chronic inflammatory autoimmune CNS disease that cause demyelination, axonal loss, and neurodegeneration.[192] MS is estimated to affect 2.5 million people worldwide and nearly 1 million people in the U.S.[193,194] The pathophysiology of MS is complex and is mainly related to two fundamental process described as (1) focal chronic inflammation inducing macroscopic plaques and BBB injury, and (2) neurodegeneration with microscopic injury in the neurons, axons and synapses.[195] Although the cause of MS is still unknown, it is believed that MS is triggered by a combination of factors such as environmental, genetic, and non-genetic factors.[196] 

Similar to chronic neuroinflammation, MS compromises the integrity and function of the BBB. Evidence suggest that dysfunctional transporters at the BBB have been implicated in MS; however, the exact mechanism is uncertain. Several studies identified an important role for ABCB1, ABCG2, ABCC1 and ABCC2 in neuroinflammatory processes underlying MS pathology.[197-200]. Kooij et al. reported a reduced vascular expression of ABCB1 in active MS lesions from MS human brain tissue.[197] Their study also showed that in vivo, the expression and function of Abcb1 at the BBB was significantly impaired in MS animal model.[198] The same group investigated the expression pattern of different ABC transporters (ABCB1, ABCG2, ABCC1 and ABCC2) in human MS lesions.[197] ABCB1 cerebrovascular expression was reduced in active and chronic inactive MS lesions.[197] In contrast, reactive primary astrocytes showed an increase in both ABCB1 and ABCC1 expression in human brain tissues of MS patients compared to control.[197] They proposed a potential new pathophysiological role for ABCB1 and ABCC1 on reactive astrocytes: they could contribute to the inflammatory process by mediating immune cell migration and aggravating inflammatory attack during MS pathogenesis.[197]

  1. Among the cytokines, vascular endothelial growth factor (VEGF), which is  upregulated  in  many  brain diseases, is known  to  promote  BBB  leakage  in  the  ischemic  brain, in brain tumors, and  in  CNS  inflammatory diseases  by  disrupting  endothelial tight  junction  By using submicrometer-resolution multiphoton fluorescence microscopy with a  longer  penetration  depth  into  brain parenchyma, this paper first sought to quantify temporal VEGF effects on BBB permeability to various-sized molecules: Shi, L., Zeng, M. and Fu, B.M. (2014), Temporal effects of vascular endothelial growth factor and 3,5‐cyclic monophosphate on blood–brain barrier solute permeability in vivo. Journal of Neuroscience Research, 92: 1678-1689. https://doi.org/10.1002/jnr.23457

Response: as suggested by the reviewer, a sentence was added to the manuscript as follows:

 As a regulatory interface between the CNS and the immune system, the BBB is not a simple physical cellular barrier, therefore it can both impact and can be impacted by the immune system at many levels.[95] On one hand, the BBB transports and secretes several cytokines and substances with neuroinflammatory properties. For instance, the vascular endothelial growth factor (VEGF) cytokine is a key mediator in BBB damage for several neurological diseases: VEGF is known to promote BBB leakage in the ischemic brain, brain tumors and CNS inflammatory diseases due to a disruption of endothelial tight junction proteins.[96-98]  On the other hand, modulation of expression and function of essential transporters (e.g. ABCB1)

  1. The authors used transporter acronyms differently all through the manuscript, such as ABCB1 & Abcb1, ABCG2 & Abcg2, SLCOs & Slcos, these better be consistent.

Response: The nomenclature corresponding to specific species is used throughout the manuscript. The nomenclature changes depending of animal species. In human (human specific or human homolog), symbols (gene or protein) are written all caps. However, symbols for rodents begin with an upper-case letter (only the first letter) followed by all lowercase letter in lowercase. When transporters refer to a rodent (mouse or rat studies), their corresponding nomenclature is used. In addition, symbols for genes are italicized whereas symbols for proteins are not.

  1. Line 185, “ABCG2” – this is Section 2.1.2, therefore, Section “2.1.2 ABCCs” (Line 206) is Section “2.1.3”.

Response: We thank the reviewer for pointing out this typo.  Sections describing ABCG2 and ABCCs are now 2.1.2 and 2.1.3, respectively.